# Analysis of the Thermal Storage Performance of a Radiant Floor Heating System with a PCM

**DOI:** 10.3390/molecules24071352

**Published:** 2019-04-05

**Authors:** JinChul Park, TaeWon Kim

**Affiliations:** 1Department of Architecture, Chung-Ang University, Seoul 156-756, Korea; jincpark@cau.ac.kr; 2Department of Architecture Engineering, Graduate School of Chung-Ang University, Seoul 156-756, Korea

**Keywords:** phase change material, radiant floor heating system, thermal storage, mock-up test

## Abstract

This study first reviewed previous studies on floor heating systems based on the installation of a phase change material (PCM) and the current status of technical developments and found that PCM-based research is still in its infancy. In particular, the improvement of floor heat storage performance in indoor environments by combining a PCM with existing floor structures has not been subject to previous research. Thus, a PCM-based radiant floor heating system that utilizes hot water as a heat source and can be used in conjunction with the widespread wet construction method can be considered novel. This study found the most suitable PCM melting temperature for the proposed PCM-based radiant floor heating system ranged from approximately 35 °C to 45 °C for a floor thickness of 70 mm and a PCM thickness of 10 mm. Mock-up test results, which aimed to assess the performance of the radiant floor heating system with and without the PCM, revealed that the PCM-based room was able to maintain a temperature that was 0.2 °C higher than that of the room without the PCM. This was due to the rise in temperature caused by the PCM’s heat storage capacity and the emission of waste heat that was otherwise lost to the underside of the hot water pipe when the PCM was not present.

## 1. Introduction

The United Nations Framework Convention on Climate Change—the international organization established to address global climate change—has targeted greenhouse gas reduction as a top priority objective in recent years. Based on a voluntary greenhouse gas reduction roadmap, a number of developed and developing nations, including those that have not yet participated in existing reduction activities, have agreed to reduce greenhouse gas emissions by 40% to 70% by 2050 and to progressively decarbonize as a means to reduce the global mean temperature by 1.5–2.0 °C [1].

In light of this, South Korea has set a 2030 target of reducing greenhouse gas emissions by 37% from business-as-usual levels, and has attempted to achieve greenhouse gas reduction by establishing new energy policies across all sectors [2]. Of particular concern are greenhouse gas emissions from buildings, which account for 23% of all greenhouse gas emissions, with greenhouse gas emissions from heating energy in particular generating 54% of these emissions in residential and nonresidential areas, respectively [3].

Most apartments housing in South Korea employ a floor heating system based on hot water and concrete slabs: this system stores heat energy in the floor using hot water supplied by a boiler and emits radiated heat from the floor surface to heat the room [4,5]. The most commonly used heat storage materials for the floor are autoclaved lightweight concrete and mortar, which are placed above and below the hot water pipes thus storing the heat and maintaining a longer heating time. However, these materials have low heat storage performance, thus a large amount of hot water and energy is typically required [6].

This study thus aims to analyze heat storage performance after the addition of a phase change material (PCM) to the conventional radiant floor heating system. First, previous research on PCM radiant floor heating systems was analyzed. A PCM-based radiant floor heating system suitable for Korean apartment housing was then designed, and the optimal range for the melting temperature of the PCM for this system was calculated. Following this, a mock-up that included a reference room and a PCM-installed room was constructed and tests were performed to verify the performance of the proposed PCM-based radiant floor heating system.

## 2. Analysis of Previous Research

A number of previous studies have proposed PCM-based radiant floor heating systems and reported the results of trials. For example, Yoon [7] proved experimentally the applicability of a new renewable energy system for floor heating, while Yoon [8] proposed a system that supplied hot water using a solar thermal and geothermal system for the heating of building floors using multidirectional valves and heat pumps. Isone Industry Co. Ltd. developed a wooden heat storage floor using a PCM and hot water pipes that was constructed by assembling a modularized PCM floor finish material [9]. Lin et al. [10] manufactured a shape-stabilized PCM (SSPCM) in plate form and designed a radiant floor heating system using night electricity. Cheng et al. assessed the thermal performance and reduction in energy consumption of a radiant floor heating system using a heat conduction enhanced, shape-stabilized PCM (HCE-SSPCM) [11]. Jin et al. [12] inserted cold and hot water pipes into concrete to act as a heating and cooling source for a PCM-based radiant floor heating system and verified the performance of this system through experiments. Zhou et al. [13] evaluated the performance of a floor heating system to which PCM and capillary electric mats had been applied based on the type of heat storage material and the type of heat source. Huang et al. [14] proposed a hybrid PCM-based radiant floor heating system that fused a PCM-based radiant floor heating system and solar thermal hot water system using new and renewable energy and assessed its thermal performance. Mitsubishi Corporation, Japan, developed a radiant floor heating system using a PCM’s latent heat for residential and office buildings [15]. Negishi Industry Co., Ltd., Japan developed a PCM-based radiant floor heating system using dry and wet construction methods and designed the system mainly for wooden houses [16]. A pilot study for a floor heating system using a PCM was conducted by ‘P’ company in the UK (PCM Ltd, Northants, UK, 2017). This was a hybrid system that combined a solar water system with a floor structure into which a PCM had been inserted via polyethylene tubes. [17].

The PCM-related research and current status of technology summarized above indicate that PCM-related theoretical and validation research is still in its early stages. In particular, research into improvements in energy performance in indoor environments by combining a PCM and an existing floor structure has not yet been conducted. South Korea typically employs wet construction methods for its floors, while other nations tend to employ dry construction. As such, PCM-based floor heating systems using an electrical heat source or those constructed by inserting a PCM into the floor structure via dry methods are dominant. In addition, no previous studies have been conducted on PCM-based systems utilizing the heat generated by hot water pipes. Even when a wet construction method has been proposed, previous research has involved a large difference in operating methods or has proposed an electrical heat source. Some countries employ natural-type or facility-type PCM radiant floor heating systems based on dry methods or electric heat sources.

The present study thus proposes a PCM-based radiant floor heating system that can be used in conjunction with the existing wet construction method. This system employs a PCM to act as a high-performance heat storage material within a conventional floor structure that only has sensible heat areas [18]. Based on the analysis above, the PCM radiant floor heating system proposed in this study, which utilizes hot water and can be used with wet construction, can be considered novel.

## 3. Design of the PCM-Based Radiant Floor Heating System and Determining the Optimal PCM Melting Temperature Range

### 3.1. Summary

Figure 1 summarizes the heat transfer dynamics of a floor heating system containing a PCM in Korean housing. If the heat supply is turned off in a hot water floor heating system without a PCM, the surface temperature cools rapidly, leading to the continuous operation of the boiler controller (Figure 2). However, the floor heating system with a PCM can keep the room warm even after the hot water supply is interrupted because the heat energy from the hot water is stored in the PCM as latent heat (Figure 3).

### 3.2. Design of the PCM-Based Radiant Floor Heating System

The PCM-based radiant floor heating system proposed in this study consisted of a 210-mm-thick concrete slab, 20-mm-thick cushioning material, 40-mm-thick autoclaved lightweight concrete, and 40-mm-thick mortar (Figure 4). This meets the existing standard floor structure thickness criteria in Korea. The PCM was placed within the autoclaved lightweight concrete layer, below the hot water pipe to store waste heat that would otherwise be lost from the pipe, thus improving overall heat storage performance within the existing autoclaved lightweight concrete and mortar layers. 

This design is also practical and convenient in that it is compatible with existing construction methods: it does not require additional processes except for the replacement of the autoclaved lightweight concrete with mortar and the insertion of the PCM. In addition, because the density of the replacement mortar and the PCM is higher than that of the autoclaved lightweight concrete, the proposed system is expected to satisfy existing standards regarding the regulation of heavy and light impact sounds.

The installation location of the PCM in the floor heating system in this study was determined by first testing two locations: next to and under the heating tube. This preliminary testing found that the latter location was better than the former because heat transfer in the hot water tube was more effective in the vertical direction than horizontally.

### 3.3. Determining the Optimal PCM Melting Temperature Range

For any PCM-based radiant floor heating system, comfortable indoor and floor surface temperatures should be established. It is thus necessary to calculate a temperature range that considers the relationship between the melting temperature of the PCM and the indoor and floor surface temperatures. This calculation first determines the heating value ${}_{PCM}Q_{indoor}$ for the indoor air from the PCM in the latent heat emitting areas, and this is then used to derive the mortar and floor surface temperatures $T_{mortar}$ and $T_{underfloor}$, respectively, which can be compared with the established conditions. Thus, a comfortable indoor and floor surface temperature should be defined for the heating period in apartments. The Design Standards of Building Energy Saving No. 2017-71 released by the Ministry of Land, Infrastructure, and Transport, Korea proposed indoor temperature and humidity criteria for calculating the capacity of cooling and heating facilities (Appendix 8). The indoor temperature criteria in this document suggested a range of 20 to 22 °C, so a high of 22 °C was selected as a comfortable indoor temperature for the proposed system.

For heat resistance ($R_{underfloor}$) by convection and radiation from the floor surface, reference values for indoor and outdoor surface heat transfer resistance were applied to the calculation of the heat transfer rate, as suggested by the energy-saving design of the building. Thus, the total heat transfer resistance on the floor surface in an apartment building (0.717 °C/W) was divided by the surface area of the floor module (0.086 m^2^·°C/W) (National Land Transport Ministry, 2017). Therefore, the total heat resistance (R) of the floor, combined with the thermal resistance of each material and the floor surface, was calculated to be 1.875 °C/W. In addition, the PCM melting temperature ($T_{PCM}$) was set manually because there was no separate calculation method available for this. Because the room temperature was set at 22 °C, the temperature of the PCM needed to be higher than the setting for the room temperature in order for heat to be transferred from the floor to the room. Based on this, $T_{PCM}$ was initially set at 23 °C, i.e., 1 °C higher than the room temperature, and this was increased in 1 °C intervals to a maximum of 52 °C (Table 1). The latent heat emission by the PCM and the surface temperature of the mortar and floor were then calculated using Equations (1)–(3), respectively.
(1)$$T_{mortar}=T_{PCM}-\left[PCMQ_{indoor}\right]\cdot R_{mortar}$$
(2)$$T_{mortar}=T_{PCM}-\left[PCMQ_{indoor}\right]\cdot R_{mortar}$$
(3)$$T_{underfloor}=T_{mortar}-\left[PCMQ_{indoor}\right]\cdot R_{finishing}$$
$R_{mortar}:$Thermal conductance resistance of the mortar (°C/W)$R_{finishing}:$Thermal conducting resistance of the finishing mortar (°C/W)$R_{underfloor}:$Convective resistance of the floor surface (°C/W)R:Total heat conductive resistance of the floor structure (°C/W)$T_{PCM}:\;$Represents the melting temperature (°C)$T_{indoor}:$Indoor temperature (°C)$PCMQ_{indoor}:$The amount of heat supplied indoors by the PCM (W)$T_{mortar}:\;$Surface temperature of the mortar (°C)$T_{underfloor}:$Surface temperature of the floor (°C)

The PCM melting temperature that generated an indoor temperature of 22 °C, and a floor surface temperature of 28–30 °C was then calculated and set at 35–45 °C with an error of ±1 °C (Figure 5). Thus, when a floor’s heat storage layer consisted of 70-mm-thick mortar and a 10-mm-thick PCM, the most suitable PCM melting temperature for the radiant floor heating system proposed in this study to produce comfortable indoor temperatures was found to be within the range of 35 to 45 °C. 

## 4. Mock-Up Testing of the PCM Floor Heating System

### 4.1. Construction of the Mock-Up Test Room

A mock-up test room was constructed indoors in accordance with the standard floor structure proposed by the Ministry of Land, Infrastructure, and Transport, Korea to compare and verify the performance of the radiant floor heating system with and without the PCM. The test room is shown in Figure 3, and a summary of the constructed mock-up is presented in Table 2. The rooms were divided into two, each of which had a floor that followed the specifications of the standard floor structure. The floor area and volume of each room were 3.6 m^2^ and 10.6 m^3^, respectively, and each room was constructed to have enough space for an individual adult to engage in regular activities. The walls of the room had a 200-mm-thick insulation layer to block thermal exchange with the outside and the external conditions (which included an external temperature of ~20 °C regulated by the artificial climate of the laboratory) were kept constant (Figure 6).

The floor was divided into four areas to improve the accuracy of the data measurement before the rooms were constructed, and temperature sensors were installed in each of these areas and the mean was calculated. The sensors were installed on the upper surface of the hot water pipe supplied to each room, on the upper surface of the PCM, on the floor panel surface, and 1200 mm above the floor surface (i.e., at the average human breathing height) to measure the air temperature. A diagram of the floor structure and the positions of the installed temperature sensors are shown in Figure 7 and Figure 8.

Table 3 presents the physical properties of the PCM applied to the floor heating system for the mock-up test.

### 4.2. Determining the Optimal Volume of Packaging for the PCM

The heating value delivered to the PCM from the hot water pipe was calculated. This was determined to be the minimum heat required for the complete melting of the PCM during the daily operation (i.e., eight hours) of the boiler. To calculate this value, the hot water pipe temperature, PCM temperature, heat storage time, and PCM heat storage were considered. The heat storage in the PCM was 165 kJ/kg based on the PCM TEST Result Report provided by the manufacturer. 

Equation (4) was calculated to determine the amount of heat transferred from the inserted hot water piping to the PCM based on the heat accumulation time and the heat accumulation of the PCM. The thermal resistance of the hot water piping was not included in the calculation due to the desire to develop a generalized calculation formula and in consideration of the diverse range of hot water piping materials available. Equation (5) was used to calculate the heat accumulation time and the amount of heat obtained using the values obtained.

The heat storage of the PCM calculated using the above method was 3376 kJ, so the optimal capacity of the PCM was determined to be 20 kg/room.
(4)$$Q=\frac{T_{pipe}-T_{PCM}}{R_{mortar}}$$
(5)$$Q\cdot H_{st}\leq PCM_{tc}$$


Q:
The amount of heat transferred from the hot water pipe to the PCM [W]
$T_{pipe}:$
Hot water pipe temperature [°C]
$T_{PCM}:$
PCM temperature [°C]
$H_{st}:$
Heat accumulation time [h]
$PCM_{tc}:$
Thermal storage capacity of PCM [kJ]
$R_{mortar}:$
Thermal resistance of the mortar [°C/W]

A PCM moves between a liquid and solid state during the phase change process. Thus, PCM leakage needed to be prevented, so the PCM was fully sealed using a vacuum pack. Polyethylene resin, aluminum sheets, or pipes are generally used to hold a PCM when it is inserted into the floor structure (Mitsubishi Ltd., Fukuoka, Japan, 2017). In this study, thin-film aluminum was chosen, which has a thermal conductivity coefficient of ~237 W/m^2^ °C, due to its adhesion to mortar, ability to withstand high pressures, low internal corrosion, and high thermal conductivity. The aluminum used is very thin (0.05 mm), thus it had little effect on the total thickness of the floor structure.

The air around the PCM was removed in the vacuum packing process, reducing the volume, and the opening of the packaging was then sealed using double hot wires at temperatures above 200 °C. Figure 9 shows the final appearance and size of the PCM packages.

### 4.3. Hot Water Boiler Operating Schedule 

The operating schedule for the boiler was set based on a typical family of four (Table 4). The boiler only operated when the residents were home and during sleeping hours. As presented in Table 5, the boiler was turned on from 18:00 until 23:00, which was one hour before sleep, and again from 3:00 to 6:00, for a total of eight hours per day. The hot water temperature was set at 55 °C in both model rooms. 

Temperature sensors were installed on the upper surface of the hot water pipe in each room, on the upper surface of the PCM, and on the floor panel surface. The sensors used in the temperature measurement were T-type thermocouples (T0.32-Y-W-15), and measurements were taken every minute using a Midi Logger GL820 from Graphtec. Table 6 details the locations of the installed temperature sensors. Only the temperature of the upper surface of the PCM was measured because the heat storage energy lost from the lower surface was considered too minimal to affect the floor surface temperature. 

## 5. Results

### 5.1. Temperature on the Upper Surface of the Hot Water Pipe

The temperature on the upper surface of the hot water pipe while the boiler was running was 0.2 °C lower on average in the PCM-containing Room 2 than in Room 1 (without PCM). However, the temperature eight hours after the boiler had been turned off was 1 °C higher in Room 2 (with PCM, 30.8 °C) than in Room 1 (without PCM, 29.8 °C). This is due to the heat storage effect of the PCM (Figure 10 and Table 7).

### 5.2. Temperature on the Upper Surface of the PCM

The temperature on the upper surface of the PCM was measured for the PCM-based Room 2. The temperature on the upper surface of the PCM rose from 26.5 °C prior to the boiler being switched on to 35.8 °C five hours later. Also, the PCM temperature remained at over 34 °C four hours after the boiler had been turned off. Furthermore, the temperature on the upper surface of the PCM then increased to 38.5 °C when the boiler was switched on again (Figure 11 and Table 8).

### 5.3. Floor Surface Temperature

The floor surface temperatures of Room 1 (without PCM) and Room 2 (with PCM) were compared; it was found that Room 2 had a consistently higher temperature than Room 1 by 0.5 °C to 0.8 °C on average. Room 2 maintained this higher temperature even after the boiler was turned off due to the effect of PCM heat storage (Figure 12 and Table 9).

### 5.4. Air Temperature Indoors

The air temperature indoors, which was measured 1200 mm above the floor surface, was on average 0.2 °C higher in the room with PCM installed (Room 2; average of 28.5 °C) than in Room 1 (without PCM). Room 2 was 0.3 °C higher on average four hours after the boiler had ceased operation (Figure 13 and Table 10).

The experimental results verified that the room with the PCM installed maintained higher temperatures than did the room without a PCM. In particular, the upper surface of the PCM maintained a temperature above 36 °C four hours after the boiler had been switched off, which was more than 2 °C higher than the other room.

This means that the floor structure that contained the PCM had greater heat storage, thus consistently maintaining a higher temperature on both the floor surface and in the air. The above results were due to the PCM storing and then emitting the waste heat that was otherwise lost from the lower surface of the hot water pipe.

## 6. Discussion and Conclusions

In this study, a PCM was applied to an existing radiant floor heating system and the heat storage performance of the PCM was analyzed. The study results can be summarized as follows. First, previous research on the technical developments in PCM-based floor heating systems was analyzed, and it was found that the improvement of floor heat storage performance in indoor environments by combining a PCM with existing floor structures had not been attempted. Thus, the present study proposed a PCM-based radiant floor heating system with hot water as a heat source that can be used in conjunction with the widespread wet construction method in Korea. This study found that the most suitable PCM melting temperature for the proposed PCM-based radiant floor heating system was 35–45 °C for a floor thickness of 70 mm and a PCM thickness of 10 mm. The mock-up test, which aimed to assess the performance of the radiant floor heating system with and without the PCM, found that the room with PCM was able to maintain a temperature 0.2 °C higher than that of the room without the PCM. This was due to the rise in temperature caused by the PCM’s heat storage capacity and the emission of waste heat that was otherwise lost to the underside of the hot water pipe in the absence of the PCM.

However, because important variables such as solar radiation through windows, indoor air flow, and the radiant heat of indoor lighting were excluded from calculations, the PCM melting temperature reported in this study may differ from the actual optimal PCM melting temperature. In addition, thin-film (0.05 mm) aluminum packaging was used to install the PCM in the floor heating structure. However, this thin-film aluminum packaging has poor thermal conductivity, so the selection of packaging materials with high thermal conductivity should be considered in the future.

The PCM used in this study was RT-44 (Rubitherm^®^), a member of the n-paraffin PCM family, which has a constant temperature of 35 to 45 °C. PCMs are manufactured as a unit module in Korea after import, and their price remains high compared to heat accumulators at around $40/L. However, if the environmental costs are considered, such as greenhouse gas production, PCM systems represent an economically viable option. As a result of this research, it is expected that the use of PCM materials in floor radiant heating systems will help to create a comfortable indoor environment by reducing dramatic fluctuations in indoor temperatures via the thermal storage effect.

## Figures and Tables

**Figure 1 molecules-24-01352-f001:**
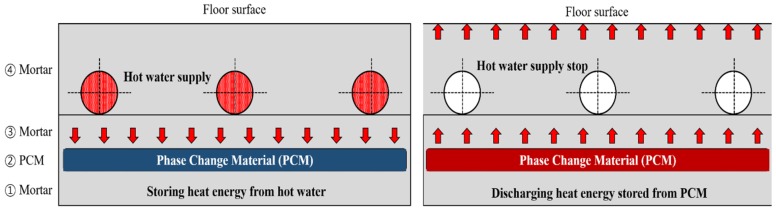
Heat emission from a floor heating system containing a phase change material (PCM).

**Figure 2 molecules-24-01352-f002:**
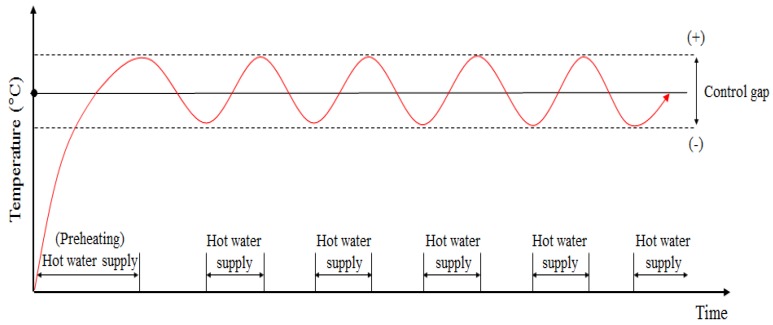
Hot water supply over time for a floor heating system without a PCM.

**Figure 3 molecules-24-01352-f003:**
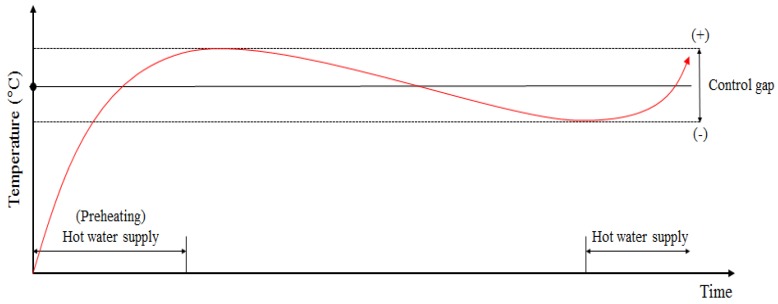
Hot water supply over time for a floor heating system with a PCM.

**Figure 4 molecules-24-01352-f004:**
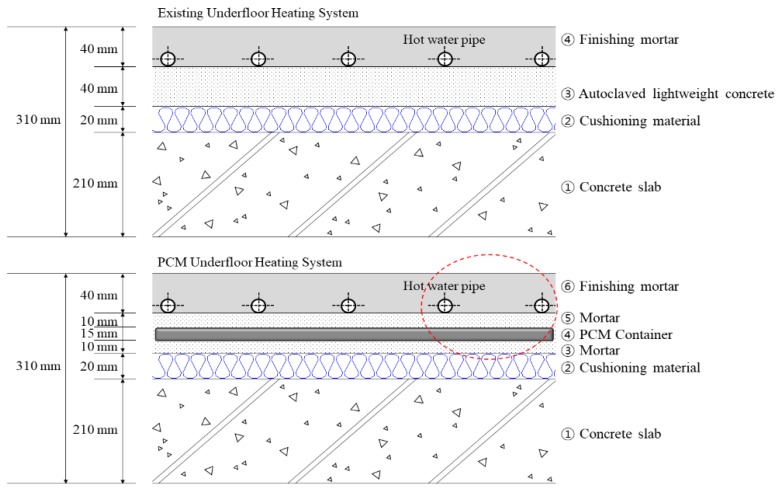
Design of the proposed PCM-based radiant floor heating system structure.

**Figure 5 molecules-24-01352-f005:**
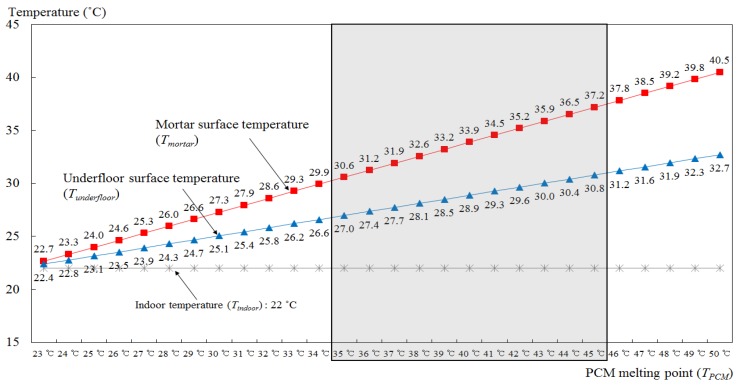
Mortar and floor surface temperatures according to changes in PCM melting temperature.

**Figure 6 molecules-24-01352-f006:**
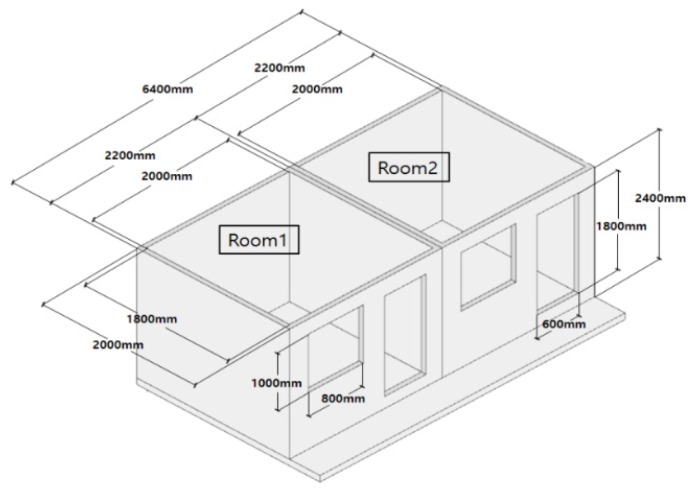
3D visualization of the mock-up rooms.

**Figure 7 molecules-24-01352-f007:**
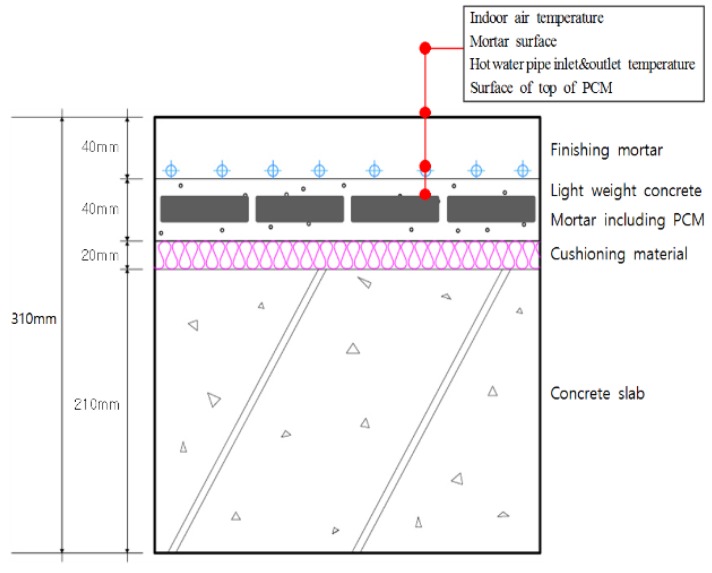
Sensor locations in the conventional radiant heating system.

**Figure 8 molecules-24-01352-f008:**
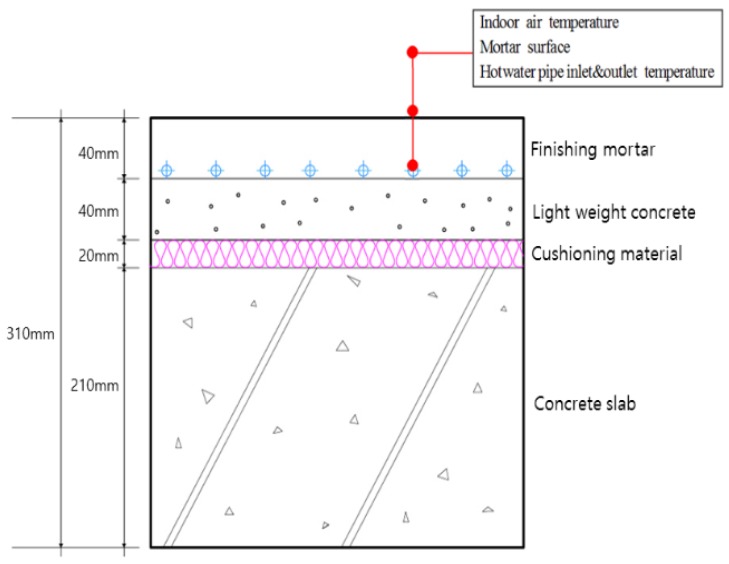
Sensor locations in the PCM-based radiant heating system.

**Figure 9 molecules-24-01352-f009:**
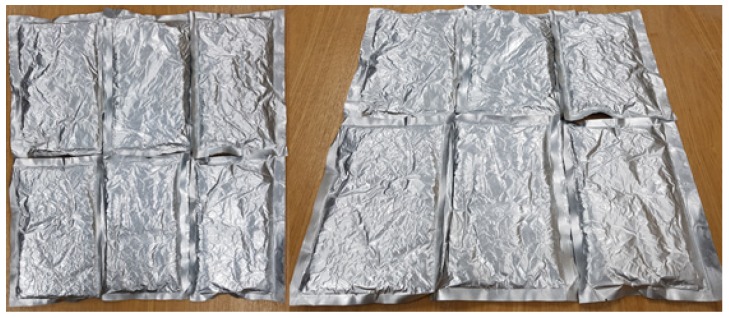
Images of the PCM packaging.

**Figure 10 molecules-24-01352-f010:**
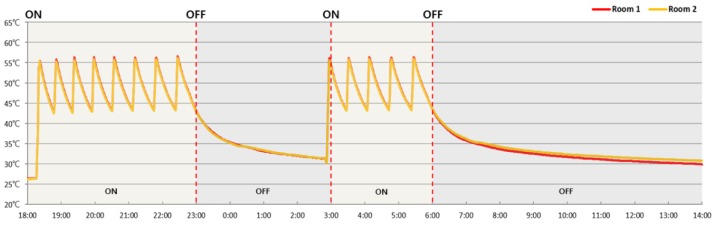
Hot water temperature on the surface of the pipe over time.

**Figure 11 molecules-24-01352-f011:**
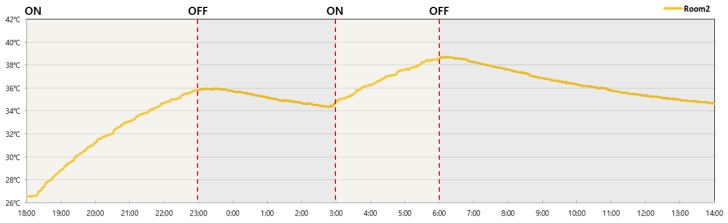
Temperature on the upper surface of the PCM over time.

**Figure 12 molecules-24-01352-f012:**
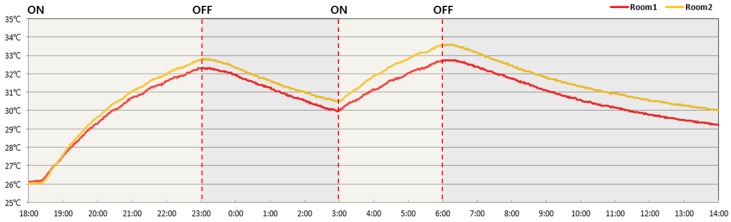
Floor surface temperature over time.

**Figure 13 molecules-24-01352-f013:**
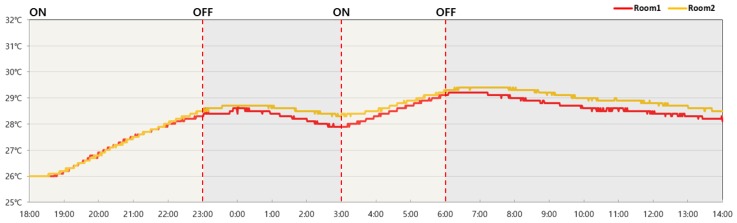
Change in room air temperature over time.

**Table 1 molecules-24-01352-t001:** Results of the calculation of PCM latent heat emission, mortar temperature, and floor surface temperature.

$R_{mortar}\;$(°C/W)	$R_{finishing}\;$(°C/W)	$R_{underfloor}\;$(°C/W)	R (°C/W)	$T_{PCM}\;$(°C)	$T_{indoor}\;$(°C)	$PCMQ_{indoor}\;$(W)	$T_{mortar}\;$(°C)	$T_{underfloor}\;$(°C)
0.637	0.521	0.717	1.875	23	22	0.5	22.7	22.4
0.637	0.521	0.717	1.875	24	22	1.1	23.3	22.8
0.637	0.521	0.717	1.875	25	22	1.6	24.0	23.1
0.637	0.521	0.717	1.875	26	22	2.1	24.6	23.5
0.637	0.521	0.717	1.875	27	22	2.7	25.3	23.9
0.637	0.521	0.717	1.875	28	22	3.2	26.0	24.3
0.637	0.521	0.717	1.875	29	22	3.7	26.6	24.7
0.637	0.521	0.717	1.875	30	22	4.3	27.3	25.1
0.637	0.521	0.717	1.875	31	22	4.8	27.9	25.4
0.637	0.521	0.717	1.875	32	22	5.3	28.6	25.8
0.637	0.521	0.717	1.875	33	22	5.9	29.3	26.2
0.637	0.521	0.717	1.875	34	22	6.4	29.9	26.6
0.637	0.521	0.717	1.875	35	22	6.9	30.6	27.0
0.637	0.521	0.717	1.875	36	22	7.5	31.2	27.4
0.637	0.521	0.717	1.875	37	22	8.0	31.9	27.7
0.637	0.521	0.717	1.875	38	22	8.5	32.6	28.1
0.637	0.521	0.717	1.875	39	22	9.1	33.2	28.5
0.637	0.521	0.717	1.875	40	22	9.6	33.9	28.9
0.637	0.521	0.717	1.875	41	22	10.1	34.5	29.3
0.637	0.521	0.717	1.875	42	22	10.7	35.2	29.6
0.637	0.521	0.717	1.875	43	22	11.2	35.9	30.0
0.637	0.521	0.717	1.875	44	22	11.7	36.5	30.4
0.637	0.521	0.717	1.875	45	22	12.3	37.2	30.8
0.637	0.521	0.717	1.875	46	22	12.8	37.8	31.2
0.637	0.521	0.717	1.875	47	22	13.3	38.5	31.6
0.637	0.521	0.717	1.875	48	22	13.9	39.2	31.9
0.637	0.521	0.717	1.875	49	22	14.4	39.8	32.3
0.637	0.521	0.717	1.875	50	22	14.9	40.5	32.7
0.637	0.521	0.717	1.875	51	22	15.5	41.1	33.1
0.637	0.521	0.717	1.875	52	22	16.0	41.8	33.5

**Table 2 molecules-24-01352-t002:** Characteristics of the model rooms used to test the proposed floor heating system.

Room Size	2.2 m × 2 m × 2.4 m (width × length × height)
**Room Volume**	10.6 m^3^
**Floor Area**	7.2 m^2^ (3.6 m^2^ per space)
**External Conditions**	Identical (indoor mock-up)
**Room Composition**	Room 1 (General): General floor heating
	Room 2 (Bottom): Base of pipe PCM (RT42)

**Table 3 molecules-24-01352-t003:** The physical properties of the PCM applied to the floor heating system.

Configuration	Details
**Components**	Proprietary blend of plant-based Kosher ingredients derived from vegetable oils such as fatty acids, fatty alcohols, fatty esters, and their derivatives and any combination of the previously mentioned products listed on the Generally Recognize as Safe (GRAS) list by the FDA that contain no petroleum or animal fat products.
**Physical and Chemical Properties**	Appearance: Colorless liquid (above melting point)
	Relative Density: 0.85–0.90 g/mL @ 45 °C
	Melting Point: 42 °C (107.6 °F)
	Boiling Point: >250 °C (482 °F)
	Solubility in Water: Insoluble
	Flash Point: >110 °C (230 °F)
	Auto-Ignition Temperature: Does not ignite

**Table 4 molecules-24-01352-t004:** Hot water boiler operation outline.

Conditions	For a Family of Four with Jobs/School		
**Timetable**	08:00–18:00 Nobody at home		
	18:00–24:00 Activity		
	24:00–08:00 Bedtime		
**Water temperature**	55 °C	**Running Time**	Twice for 8 h in total

**Table 5 molecules-24-01352-t005:** Hot water boiler operating schedule.

Time	9 h	10 h	11 h	12 h	13 h	14 h	15 h	16 h	17 h	18 h	19 h	20 h	21 h	22 h	23 h	24 h	1 h	2 h	3 h	4 h	5 h	6 h	7 h	8 h
**Operation**	OFF									ON					OFF				ON			OFF		

**Table 6 molecules-24-01352-t006:** Temperature sensor position for each layer.

Temperature Sensor Position			
**Room 1**	Hot water pipe	Floor panel surface	Indoors (1200 mm)
**Room 2**	Hot water pipe	Floor panel surface	Indoors (1200 mm) PCM (42 °C)

**Table 7 molecules-24-01352-t007:** Temperature on the upper surface of the hot water pipe.

Room 1	Room 2	Time	Operation
26.4 °C	26.3 °C	18:00	ON
50.5 °C	50.2 °C	19:00	
55.1 °C	54.7 °C	20:00	
45.0 °C	44.9 °C	21:00	
50.5 °C	50.4 °C	22:00	
43.1 °C	42.9 °C	23:00	OFF
35.3 °C	35.3 °C	24:00	
33.1 °C	33.4 °C	1:00	
31.9 °C	32.2 °C	2:00	
54.3 °C	54.0 °C	3:00	ON
44.2 °C	44.1 °C	4:00	
50.0 °C	49.8 °C	5:00	
43.5 °C	43.6 °C	6:00	OFF
35.8 °C	36.2 °C	7:00	
33.5 °C	34.3 °C	8:00	
32.5 °C	33.1 °C	9:00	
31.7 °C	32.3 °C	10:00	
31.1 °C	31.9 °C	11:00	
30.6 °C	31.5 °C	12:00	
30.2 °C	31.2 °C	13:00	
29.8 °C	30.8 °C	14:00	

**Table 8 molecules-24-01352-t008:** Temperature on the upper surface of the PCM.

Room 2	Time	Operation
26.5 °C	18:00	ON
28.8 °C	19:00	
31.2 °C	20:00	
33.0 °C	21:00	
34.7 °C	22:00	
35.8 °C	23:00	OFF
35.7 °C	24:00	
35.1 °C	1:00	
34.7 °C	2:00	
34.7 °C	3:00	ON
36.2 °C	4:00	
37.6 °C	5:00	
38.5 °C	6:00	OFF
38.2 °C	7:00	
37.6 °C	8:00	
36.8 °C	9:00	
36.3 °C	10:00	
35.7 °C	11:00	
35.3 °C	12:00	
35.0 °C	13:00	
34.6 °C	14:00	

**Table 9 molecules-24-01352-t009:** Temperature on the floor surface.

Room 1	Room 2	Time	Operation
26.1 °C	26.0 °C	18:00	ON
27.5 °C	27.5 °C	19:00	
29.2 °C	29.6 °C	20:00	
30.6 °C	31.0 °C	21:00	
31.6 °C	31.9 °C	22:00	
32.2 °C	32.7 °C	23:00	OFF
31.9 °C	32.3 °C	24:00	
31.2 °C	31.6 °C	1:00	
30.5 °C	31.0 °C	2:00	
29.9 °C	30.5 °C	3:00	ON
31.1 °C	31.8 °C	4:00	
32.0 °C	32.8 °C	5:00	
32.7 °C	33.5 °C	6:00	OFF
32.3 °C	33.1 °C	7:00	
31.7 °C	32.4 °C	8:00	
31.1 °C	31.8 °C	9:00	
30.5 °C	31.3 °C	10:00	
30.1 °C	30.9 °C	11:00	
29.7 °C	30.5 °C	12:00	
29.5 °C	30.3 °C	13:00	
29.1 °C	29.9 °C	14:00	

**Table 10 molecules-24-01352-t010:** Temperature of the air indoors.

Room 1	Room 2	Time	Operation
26.0 °C	26.0 °C	18:00	ON
26.3 °C	26.2 °C	19:00	
26.9 °C	26.8 °C	20:00	
27.5 °C	27.5 °C	21:00	
28.0 °C	28.0 °C	22:00	
28.3 °C	28.5 °C	23:00	OFF
28.4 °C	28.7 °C	24:00	
28.4 °C	28.7 °C	1:00	
28.2 °C	28.5 °C	2:00	
27.9 °C	28.3 °C	3:00	ON
28.3 °C	28.5 °C	4:00	
28.7 °C	28.9 °C	5:00	
29.1 °C	29.3 °C	6:00	OFF
29.2 °C	29.4 °C	7:00	
29.0 °C	29.3 °C	8:00	
28.8 °C	29.2 °C	9:00	
28.6 °C	29.0 °C	10:00	
28.5 °C	28.9 °C	11:00	
28.4 °C	28.8 °C	12:00	
28.3 °C	28.7 °C	13:00	
28.1 °C	28.5 °C	14:00	

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
