# Peer review of "Analysis of the Thermal Storage Performance of a Radiant Floor Heating System with a PCM"

_molecules, 2019, doi:10.3390/molecules24071352_

Round 1

Reviewer 1 Report

Most of the references are Korean or Japanese, so I suggest enlarging the reference section adding also other international papers. If possible, I ask to report also the main results of the cited documents, not only their topic.

Did the Authors try to put the PCM in a different position inside the floor system, for example, over the hot water pipes?

How did the Authors calculated the results of Fig. 5?

Referring to line 159, which is the external temperature?

I also ask to better explain the assumptions at lines 183-187 (page 7).

In the discussion, at line 288: the right value is 0.2°C, not 2°C.

As a last remark, do the Authors think that the small improvement in the room air temperature can justify the higher cost of the new solution? Can they estimate the increase in cost? Probably, the room temperature increase is also related to the external temperature: is it possible estimating a value for the annual average improvement?

Author Response

Thank you for your careful review.

Reviewer 2 Report

This paper investigates the thermal storage of floors. The concept is somewhat efficient innovative.

The authors are encouraged to discuss the performance of the PCM material under fatigue and its interaction with the host material.

Author Response

The modifications are at the end of the file.

Thank you for your meticulous review.

Round 2

Reviewer 1 Report

The Authors have enough satisfactorily answered to the requirements of the reviewers.